# Mapping QTLs underpin nutrition components in aromatic rice germplasm

M. Z. Islam[1]*, M. Arifuzzaman[2], S. Banik[3], M. A. Hossain[4], J. Ferdous[5], M. Khalequzzaman[1], B. R. Pittendrigh[6], M. Tomita[7]*, M. P. Ali[8]*

1 Genetic Resources and Seed Division, Bangladesh Rice Research Institute (BRRI), Gazipur, Bangladesh, 2 Department of Genetics and Plant Breeding, Hajee Mohammad Danesh Science and Technology University, Dinajpur, Bangladesh, 3 Grain Quality and Nutrition Division, Bangladesh Rice Research Institute (BRRI), Gazipur, Bangladesh, 4 Regional Station, Barisal, Bangladesh Rice Research Institute, Gazipur, Bangladesh, 5 Biotechnology Division, Bangladesh Rice Research Institute (BRRI), Gazipur, Bangladesh, 6 Department of Entomology, Michigan State University, East Lansing, Michigan, United States of America, 7 Genetics and Genome Engineering Laboratory, Green Biology Research Division, Research Institute of Green Science and Technology, Shizuoka University, Shizuoka City, Shizuoka, Japan, 8 Entomolgy Division, Bangladesh Rice Research Institute (BRRI), Gazipur, Bangladesh

* zahid.grs@gmail.com (MZI); panna_ali@yahoo.com (MPA); tomita.motonori@shizuoka.ac.jp (MT)

**Data Availability Statement:** The data underlying the results presented in the study are available within manuscript.

## Abstract

As rice is an important staple food globally, research for development and enhancement of its nutritional value it is an imperative task. Identification of nutrient enriched rice germplasm and exploiting them for breeding programme is the easiest way to develop better quality rice. In this study, we analyzed 113 aromatic rice germplasm in order to identify quantitative trait loci (QTL) underpinning nutrition components and determined by measuring the normal frequency distribution for Fe, Zn, amylose, and protein content in those rice germplasm. Comparatively, the germplasm Radhuni pagal, Kalobakri, Thakurbhog (26.6 ppm) and Hati-sail exhibited the highest mean values for Fe (16.9 ppm), Zn (34.1 ppm), amylose (26.6 ppm) and protein content (11.0 ppm), respectively. Moreover, a significant linear relationship ($R^2 = 0.693$) was observed between Fe and Zn contents. Cluster analysis based on Mahalanobis $D^2$ distances revealed four major clusters of 113 rice germplasm, with cluster III containing a maximum 37 germplasm and a maximum inter-cluster distance between clusters III and IV. The 45 polymorphic SSRs and four trait associations exhibited eight significant quantitative trait loci (QTL) located on eight different chromosomes using composite interval mapping (CIM). The highly significant QTL (variance 7.89%, LOD 2.02) for protein content (*QTL.pro.1*) was observed on chromosome 1 at 94.9cM position. Also, four QTLs for amylose content were observed with the highly significant *QTL.amy.8* located on chromosome 8 exhibiting 7.2% variance with LOD 1.83. Only one QTL (*QTL.Fe.9*) for Fe content was located on chromosome 9 (LOD 1.24), and two (*QTL.Zn.4* and *QTL.Zn.5*) for Zn on chromosome 4 (LOD 1.71) and 5 (LOD 1.18), respectively. Overall, germplasm from clusters III and IV might offer higher heterotic response with the identified QTLs playing a significant role in any rice biofortification breeding program and released with development of new varieties.

**Funding:** National Agricultural Technology Project (NATP): Phase-1 in Bangladesh. Financed by Bangladesh Agricultural Research Council (BARC), Farmgate, Dhaka. Grant No. 12. The funders had no role in study design, data collection and analysis, decision to publish, or preparation of the manuscript.". Lead author MZ Islam received fellowship from this grant. M. Khalequzzaman received project award". The funders had no role in study design, data collection and analysis, decision to publish, or preparation of the manuscript.

**Competing interests:** The authors have declared that no competing interests exist.

# 1. Introduction

While nutritious crop varieties (including those enriched with micronutrients) represent a crucial demand for healthy growth and development of human beings, few micronutrient-enriched food crop varieties exist among global staple cultivated crop varieties. In particular, iron (Fe) is an essential element for hemoglobin with zinc (Zn) also providing a key co-factor for more than three hundred enzymes involved in principal biological activities [1, 2]. Deficiencies of Fe and Zn cause a wide range of health problems, including but not limited to anemia, reduced growth, poor cognitive development, stunting, reduced immunity, diarrhea, lesions on eyes and skin, delayed healing of wounds, and mental lethargy [3, 4]. Currently, more than two billion people in the world—especially children and pregnant or lactating women—suffer from Fe and Zn deficiency diseases [5]. And while micronutrient enriched supplementary drugs are available to remediate these problems, such drugs are typically prescribed by doctors only after micronutrient deficiency symptoms appear and are often prohibitively expensive [6].

According to FAO, more than 90% of Asian people depend on rice as food, delivering micronutrients via rice would afford these missing micronutrient benefits in a more cost-effective and wider-reaching way [7, 8]. "Biofortification"—or, the hybridization of food crops with rich vitamin and mineral densities—represents one powerful and cost-effective tool for providing these micronutrients and for reducing or eliminating micronutrient-related malnutrition [7–9]. In general, although consumers prefer milled rice, which unfortunately provides an only limited source of Fe and Zn micronutrients, research to identify rice germplasm with higher micronutrient content can make rice biofortification breeding programs possible [10, 11]. A biofortification breeding program was succeeded for Zn-enriched rice varieties in Bangladesh, India, and Philippines [12]. However, amylose content in rice is also an important determinant for maintaining the eating and cooking quality of rice, as determined by three principal physicochemical properties: amylose content (AC), gel consistency (GC), and gelatinization temperature (GT). In general, rice growers prefer high amylose content rice, such that amylose-categorized rice germplasm would be required for any rice biofortification program to match market viability and grower and consumer preferences. Protein is an essential component of diet which provides basic function in nutrition and supply adequate amounts of needed amino acids in the body. Therefore, enhancing protein content in rice is an important criterion for consumer preference for ensuring healthy life [13].

Micronutrient Fe, Zn, amylose, and protein content varies widely among rice germplasm, landraces, cultivars, breeding lines, and wild rice. Among these, aromatic rice germplasms constitute a small but important group of rice genotypes that are popular (and increasingly in demand) in many countries of the world for their aroma and/or super-fine grain quality [14]. As such, aromatic rice represents a strategic choice for biofortification and is also readily available and popular in Bangladesh, where it comprises short and medium bold types with a mild to strong aroma [15, 16]. Although genetic diversity and population structure analysis based on phenotypic and genotypic characteristics of aromatic rice have been reported [17, 18], the micronutrient content for Fe, Zn, amylose, and protein content have not yet been analyzed.

The rice varieties enriched with micronutrients like iron (Fe), zinc (Zn), and/or other vitamins are considered better quality rice. Recently, demands for this type of rice has increased as a way to offset micronutrient deficiencies both in children and older people. While Fe- and Zn-rich rice varieties have been developed through conventional breeding approaches and cultivation in rice-growing countries, several issues—like complex genetic backgrounds, sophisticated phenotyping, lack of knowledge about molecular markers, and interactions between germplasm and environmental [19, 20]—make such an approach time-consuming to meet

growing demand and climate change effects. In contrast, identification of quantitative trait loci (QTLs) and genes for high Fe, Zn, amylose, and protein content could lead to more results more quickly. Furthermore, these genes or loci could be introgressed into cultivar backgrounds precisely through molecular marker assisted breeding (MAB) and genetic engineering approaches to enable early release of rice varieties [21].

QTL is a locus, or a part, of DNA estimated from quantitative trait and molecular marker data in a population. To map QTLs or genes, bi-parental mapping and genome-wide association studies (GWAS) have been applied to a number of rice traits [20–24]. However, GWAS have several merits over bi-parental mapping, including high resolution and scanning for numerous and rare alleles [25]. GWAS have been utilized to identify QTLs for Fe, Zn, and several other mineral elements in rice seeds [20, 24]. In addition, 31 putative QTLs have been identified for Fe and Zn (as well as Mn, Cu, Ca, Mg, P, and K) contents [26]. In one study, 14 QTLs for Fe and Zn and candidate genes *OsYSL1* and *OsMTP1* for Fe and *OsARD2*, *OsIRT1*, *OsNAS1*, and *OsNAS2* for Zn were identified in rice seeds [27].

While mapping of QTLs for Fe and Zn concentrations in non-aromatic rice has been performed, limited studies have investigated Fe and Zn concentrations (and amylose and protein content) in aromatic rice. The purpose of this study, therefore, was: (1) to identify high yield genotypes for Fe, Zn, amylose, and protein content in 113 aromatic rice germplasm, (2) to cluster the genotypes based on the studied traits, and (3) to identify QTLs for Fe, Zn, amylose, and protein content using molecular marker-trait associations studies.

## 2. Results

### 2.1 Mean comparison and frequency distribution

Fig 1 and Table 1 present frequency distributions and means comparison of 113 aromatic germplasm for four phenotypic traits (Fe, Zn, amylose, and protein content). Fig 1 depicts a normal frequency distribution, along with a wider phenotypic variation, for grain-Fe content (8.90 ± 3.72 ppm, range 1.10–16.90). The highest number of genotypes (25) occupied the 10–11 range followed by 11–12. The Zn content also showed a normal frequency distribution with an average 19.93 ± 6.48 ppm, with the highest number of genotypes (22) occupying the 20–21.66 range followed by 10.00–13.33 and 21.67–25. Overall, the Radhuni pagal genotype exhibited the maximum Zn content and the Luina genotype the minimum.

Amylose content exhibited a wider range, from 18.20–26.60, with an average for the 113 genotypes of 21.88 ± 1.91. The highest number of genotypes (17) occupied the 20.00–21.25 range followed by 21.25–22.5, with the genotype Thakurbhog exhibited the maximum amylose content and the genotype Jirabuti the minimum. The least significant difference (LSD) (5%) was estimated at 0.35 considering all genotypes for the trait.

Protein content also showed a normal frequency distribution, with an average for the 113 genotypes of 8.19±0.10 ppm. The highest number of genotypes (22) occupied the 8.00–8.5 range, with genotype Hatisail exhibiting the maximum protein content (11.00) and Sadagura (Sl -104) the minimum (6.30). The Zn content ranged from 7.30 to 34.10 with an average of 19.93 ± 0.61ppm, and the genotype Kalobakri exhibiting the maximum.

### 2.2 Regression analysis

The response function for the relationship between Zn concentration and Fe concentration (shown in Fig 2) exhibited a strong, highly significant linear relationship ($R^2 = 0.693$) between both. Zn content (ppm) was positively affected by Fe concentration and showed a strong relationship with Fe (F = 250.73, $p < 0.001$), e.g., with increasing Zn content, the Fe content increases and vice-versa.

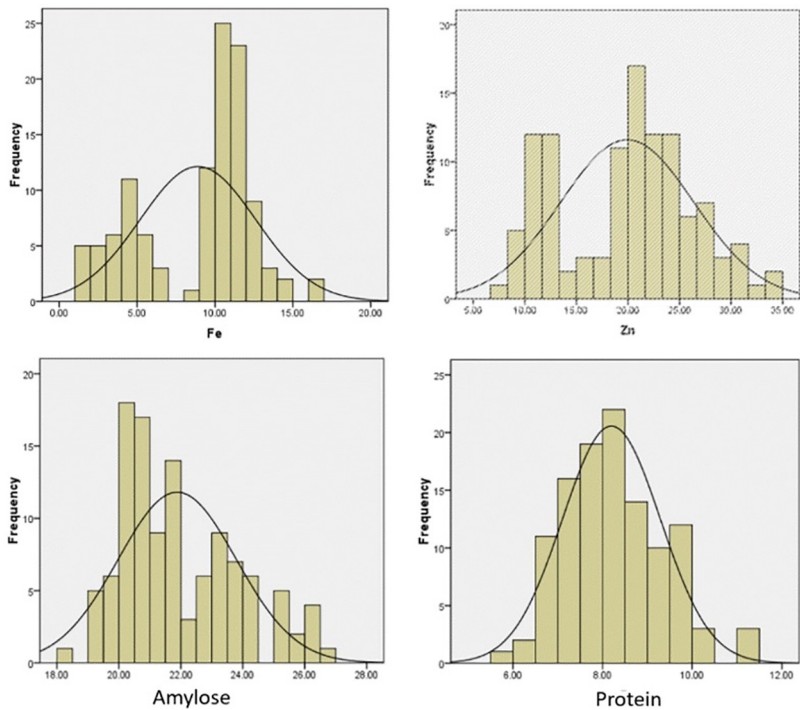

**Fig 1. Frequency distribution of 113 aromatic rice germplasm for four traits.**

Table 1. Mean values of selected superior and inferior lines for studied traits.

| Variety/ Line | Amylose | Protein | Zn | Fe |
|---|---|---|---|---|
| **Selected seven superior genotypes** | | | | |
| Kalobakri | 23.80 | 8.22 | 34.10 | 12.70 |
| Sakkorkhana | 20.70 | 9.50 | 34.00 | 11.00 |
| Hatisail | 20.40 | 11.00 | 32.60 | 11.60 |
| Rajbut | 20.60 | 9.10 | 27.80 | 10.80 |
| Sakkor khora | 20.30 | 8.00 | 27.70 | 13.50 |
| Radhuni pagal | 21.00 | 9.00 | 27.00 | 16.90 |
| Thakurbhog | 26.60 | 7.20 | 19.00 | 12.00 |
| **Selected seven inferior genotypes** | | | | |
| Lal Soru | 20.10 | 7.60 | 7.30 | 5.80 |
| Luina | 22.50 | 8.10 | 8.90 | 1.10 |
| Sadagura (Sl -104) | 24.30 | 6.30 | 18.20 | 10.30 |
| Gobindhabhog (Sl -110) | 20.80 | 6.40 | 20.60 | 9.00 |
| Jirabuti | 18.20 | 8.40 | 25.30 | 10.80 |
| Begunmala | 19.10 | 8.10 | 26.40 | 11.80 |
| Chinniguri | 19.10 | 8.30 | 20.30 | 10.20 |
| **Calculations** | | | | |
| Max. | 26.60 | 11.00 | 34.10 | 16.90 |
| Min. | 18.20 | 6.30 | 7.30 | 1.10 |
| Average | 21.88 | 8.19 | 19.93 | 8.90 |
| Std | 1.91 | 1.03 | 6.48 | 3.72 |
| SE | 0.18 | 0.10 | 0.61 | 0.35 |
| CV | 8.72 | 12.54 | 32.50 | 41.74 |
| LSD (5%) | 0.35 | 0.19 | 1.19 | 0.69 |

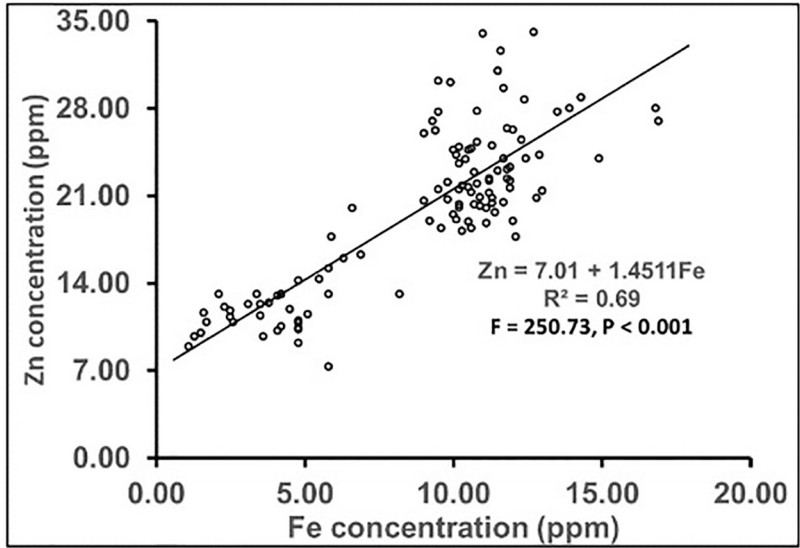

**Fig 2. Linear regression showing Zn concentration as a dependent variable on Fe concentration.**

## 2.3 Cluster analysis

The 113 germplasm were grouped into four major clusters (I, II, III and IV) at a 26% level of genetic similarity coefficient (Fig 3). Cluster I comprised 35 germplasm and was further classified into two sub-clusters IA and IB with 15 and 20 germplasm, respectively. Similarly, Cluster II comprised 21 genotypes and was further subdivided, with sub-clusters IIA and IIB having 12 and 9 germplasm, respectively, as well as higher diversity. Cluster III (with 37 genotypes, divided into IIIA and IIIB with 20 and 17, respectively) had a higher number of genotypes exhibit variable genetic distances among the germplasm. Cluster IV (with 19 genotypes, and 17 and 2 sub-clusters in IVA and IVB, respectively) was the smallest; in IVB, the two genotypes were Sakor and Chini Sagar. Higher inter-cluster was revealed between clusters III and IV, followed by clusters I and II (see Fig 3).

## 2.4 Molecular marker-trait associations

In the study, 45 polymorphic SSR markers were used in 113 rice genotypes for four phenotypic traits. Table 2 shows results of significant marker-trait associations. A total of eight significant

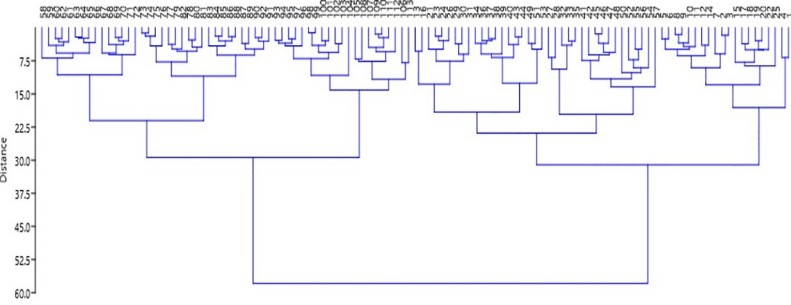

**Fig 3. A UPGMA cluster dendrogram of 113 aromatic rice germplasm based on 4 physio-chemical traits using Ward's method.**

**Table 2. Significant marker-trait associations indicating quantitative trait loci for Fe, Zn, amylose and protein content in 113 rice genotypes.**

| Traits | QTL name | Marker | Chromosome | Position (cM) | LOD | $R^2$ (%) | p-value |
|---|---|---|---|---|---|---|---|
| Fe | QTL.Fe.9 | RM215 | 9 | 1.8 | 1.24 | 3.65 | 0.0424 |
| Zn | QTL.Zn.4 | RM551 | 4 | 8.5 | 1.71 | 5.89 | 0.0095 |
| | QTL.5n.5 | RM413 | 5 | 26.7 | 1.18 | 4.54 | 0.0234 |
| Amylose | QTL.amy.6 | RM190 | 6 | 7.4 | 1.11 | 4.41 | 0.0256 |
| | QTL.amy.7 | RM125 | 7 | 63.5 | 1.45 | 5.7 | 0.0109 |
| | QTL.amy.8 | RM284 | 8 | 83.7 | 1.83 | 7.2 | 0.0041 |
| | QTL.amy.11 | RM144 | 11 | 68.6 | 1.49 | 5.9 | 0.0095 |
| Protein | QTL.pro.1 | RM5 | 1 | 94.9 | 2.02 | 7.89 | 0.0026 |

markers were identified for four nutritional traits including high Fe, Zn, amylose, and protein, with each significant marker designated a quantitative trait loci (QTL): four QTLs for protein, two QTLs for Zn, one QTL for Fe, and one QTL for protein. These QTLs were mapped using Mapdisto Version 2 (see Fig 4).

For Fe content, its single QTL was located on chromosome 9 at position 1.8cM, with 3.65% variance (p < 0.05, LOD 1.24). For Zn content, two QTLs were located on chromosomes 4 and 5 at positions 8.5 and 26.7cM, respectively, with *QTL.Zn.4* contributing higher variance (5.89%, p < 0.01 and LOD 1.71). For protein content, one QTL, *QTL.pro.1*, located on chromosome 1 at position 94.9 cM contributed a maximum 7.89% variance (p < 0.01, LOD 2.02). For amylose, 4 QTLs (*QTL.amy.6*, *QTL.amy.7*, *QTL.amy.8* and *QTL.amy.11*) were identified. In particular, *QTL.amy.8* revealed on chromosome 8 exhibited 7.2% variance (p < 0.01, LOD 1.83), followed by *QTL.amy.11* (LOD 1.49), *QTL.amy.7* (LOD 1.45) *and QTL.amy.6* (LOD 1.11).

## 3. Discussion

Plant breeding programs for biofortification of staple food crops (like rice or wheat) require screening of germplasm and varieties and/or elite lines having Fe, Zn, protein, and amylose dense grains to be used as donor parents [28]. An increase in concentration of these elements in grain is a high-priority research area. For example, maximum micronutrients are frequently present in some landraces and /or genetically distant wild varieties [29]. As such, exploitation of large genetic variation for Fe, Zn, amylose, and protein existing in cereal germplasm

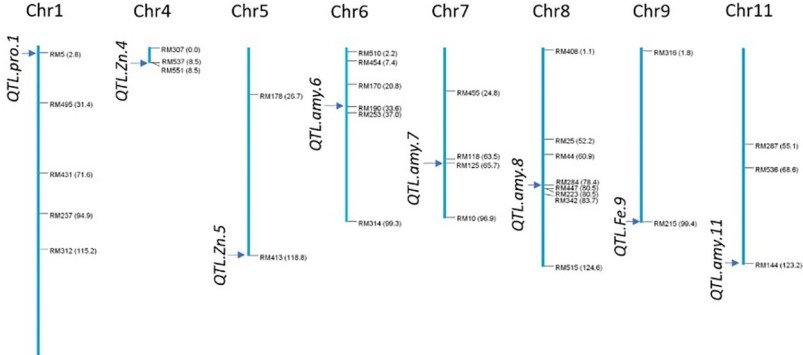

**Fig 4. QTL map showing eight significant markers for four phenotypic traits in 113 aromatic rice germplasm.**

represents an important strategy for minimizing the extent of Fe and Zn deficiencies in developing world.

In our study, the frequency distribution was normal for Fe, Zn, amylose, and protein content with maxima found in the Radhuni pagal, Kalobakri, Hatisail, and Thakurbhog genotypes, respectively. While the reasons for variably high-content varieties remain to be determined exactly, similar findings have been reported in other varieties of rice [30, 31]. One study obtained normal frequency distribution for Zn content and skewed distribution for Fe content in a recombinant inbred lines (RILs) population obtained from Madhukar×Swarna [27]. That study identified higher concentrations of Fe and Zn in Swarna (22.5ppm) and Madhukar (53.7 ppm), respectively. Moreover, pH, organic matter content, inherent Fe/Zn levels of native soil, environment, genotype, and genotype×environment interaction have all shown significant effects on rice grain Fe and Zn content [30, 31].

For genetic improvement of aromatic rice accessions in Bangladesh, a broader genetic base is required. Cluster analysis, which indicates the diversity and distances among experimental germplasm, offers a key tool in this effort. In this study, the 113 germplasms were grouped into four distinct clusters based on Mahalanobis $D^2$ distances. Previous research clustered 113 aromatic and fine rice genotypes into ten groups based on 19 quantitative traits [17], ten clusters in rice genotypes [32, 33], and clusters of 58 rice varieties in groups based on 18 morphological traits with a genetic distance of approximately 0.75 [34].

In this study, cluster III consisted of 37 genotypes, with a higher number of genotypes exhibiting variable genetic distances. While the other clusters also exhibited variable genetic distances, the highest inter-cluster was observed between III and IV. Genotypes having distant clusters offer the best opportunity for hybridization in order to achieve a higher heterotic response.

The eight significant marker trait associations (MTA) were assayed using a general linear model (GLM). Most of these loci were similar to the major loci reported for these traits [35–42]. In the genotypic analysis, seven out of the 119 SSR markers used were linked to known major-effect drought grain yield QTLs identified at International Rice Research Institute (IRRI) [38]. Here, we identified several novel loci for Fe, Zn, amylose, and protein content in rice.

The most significant QTL protein content was *QTL.pro.1* located on chromosome 1 in position 94.9 cM with LOD 2.02 and 7.89% variance. This QTL is useful for marker assisted breeding (MAB) for higher protein content in rice. Several consistent minor-effect QTLs were also identified for amylose content, with the most significant QTL located on chromosome 8. QTLs are considered significant when above the threshold LOD score 3.0 [43]. Other research reported two QTLs on chromosomes 1 (markers RM8111-RM14323, LOD 2.03) and 9 (markers RM219-RM23914, LOD 2.88) for amylose content [44], as well three QTLs on chromosomes 8 (RM506-RM1235, LOD 2.57), 9 (RM219-RM23914, LOD 2.66) and 10 (RM24934-RM25128, LOD 6.13) for protein content in 120 doubled haploid (DH) rice population. For nutrient content itself, three QTLs were identified (one for Fe and two for Zn content) [44]. In particular, the highly significant *QTL.Zn.4* located on short arm of chromosome 4 with LOD 1.71 contributed a considerable amount of variability ($R^2$ = 5.89%) [44].

Other studies obtained 42 and 3 SSR markers association with grain Fe and Zn content, respectively. Using a GLM model with $r^2$ > 0.10 and $p \leq 0.05$ filters, they reported novel QTLs *qFe3.3* and *qFe7.3* for grain Fe and *qZn2.2*, *qZn8.3* and *qZn12.3* for Zn in 485 germplasm lines of milled rice [45]. Also, one study reported QTL *qSDW3*, associated with stem dry weight and a significant LOD score of 10.7, that explained approximately 7.5% of the phenotypic variation [46], as well as another two QTLs located on chromosomes 3 and 4 with LOD scores 5.3 and 4.8 explaining 4.5% and 12.5% variance. In the same study, another locus between

RM119-RM518 was also observed on chromosome 4 that shared about 2.45% in phenotypic variance [46]

Overall, these results demonstrate that association mapping represents a feasible option for identifying major-effect QTLs for traits studied. Indeed, genome-wide association analysis (GWAS) for complex traits using molecular markers like simple sequence repeats (SSRs) are becoming a more efficient tool for identifying loci or genes for a particular trait in crop plants. In the present study, using this technique, we have identified potential candidate QTLs for bio-fortification of rice for four key micronutrient traits.

## 4. Materials and methods

### 4.1 Plant materials

In this study, 113 aromatic rice germplasm representing landraces, fine rice genotypes, elite cultivars, and exotic genotypes preserved in the genebank of Bangladesh Rice Research Institute (BRRI), Gazipur, Bangladesh were used. Names for the 113 aromatic rice germplasm, along with quantitative phenotypic traits, have been previously described [17]. Amylose and protein content were measured at the BRRI, Grain Quality and Nutrition Division, while micronutrient Fe and Zn were determined at the BRRI, Plant Breeding Division.

### 4.2 Measurement of protein content

The Micro-Kjeldahl method was used for the determination of nitrogen [47]. Fifty milligrams of powdered sample were introduced into a 30 ml Kjeldahl flask. The catalyst mixture ($K_2SO_4$, $CuSO_4$ and Selenium powder) of 1.95 g was added followed by 2.3 ml concentrated $H_2SO_4$. Digestion was continued until the mixture became clear. After digestion, the flask was connected to a distillation set up. An Erlenmeyer flask (125 ml) containing 10 ml boric acid solution plus one drop of mixed indicator was placed under the condenser with the tip of the condenser extending below the surface of the solution. Then 9 ml of $NaOH$-$Na_2S_2O_3$ solution was added slowly to the digest. The flask was connected to the steam source and distilled until about 30 ml distillate was collected. The distillate was immediately titrated against a standard HCl solution to the first appearance of a violet or reddish color. A blank determination was made simultaneously.

$$\% \ N = \frac{ml \ HCl \ for \ sample - mL \ HCl \ for \ bank}{weight \ of \ sample \ (g)} \times N \ HCl \times 0.014 \times 100$$

$$Crude \ protein \ (\%) = \% \ N \times 5.95$$

### 4.3 Measurement of amylose content

Juliano's method was used for the analysis of amylose content [48]. A milled rice sample of 100 mg was placed in a 100 ml volumetric flask. One ml of 95% ethanol and 9 ml of 1N NaOH were added to it. The contents were heated on a boiling water bath to gelatinize the starch. After cooling for one-hour, distilled water was added and the contents were mixed well. Five ml of the starch solution was put in a 100 ml volumetric flask. One ml of 1N acetic acid and 2 ml of iodine solution were added. Then distilled water was added to bring the volume up to the mark. The sample solution was set at 620 nm with a spectrophotometer. Absorbance values were plotted and a calculation was made with the help of a standard curve, available for rice samples of predetermined amylose content. Results were expressed as per cent amylose content in milled rice weight.

### 4.4 Measurement of Fe and Zn content (mg/kg)

Polished grain was analyzed by Inductively Coupled Plasma (ICP) from the Plant Breeding Division Laboratory at BRRI. For each treatment, 0.6 g of powder were weighed accurately and placed in high-pressure digestion vessels (125 ml Erlenmeyer flask). A 10 ml mixture of $HClO_4$:$HNO_3$ (1:10) was added to each flask and covered loosely with a plastic film label "perchloric" and kept overnight or longer in the fume hood. Pre-digested samples were mixed well by swirling and placed onto a hotplate and the temperature increased gradually to 225 ˚C. Then 2 ml of 2:5 mixture of $HClO_4$:$HNO_3$ were added to each sample. When digests were clear to light yellow, the temperature was increased up to 240 ˚C and heated until 1 ml of digest (sample) was left. After cooling, the samples were diluted in 10 ml of 0.5N HCl. The digested samples were used for the quantification of Fe and Zn by inductively coupled plasma-optical emission spectrophotometer (ICP-OES) using a modified procedure [49]. The concentrations of Fe and Zn were determined and presented as mg/kg (ppm). The seed-Fe and Zn contents of germplasm were estimated, and statistical analysis done using their average values.

### 4.5 Molecular characterization

We used 52 well-distributed SSRs for the molecular characterization (S1 Data). The cM positions, repeat motifs, and chromosomal positions for the SSR markers can be found in the rice genome database [50]. Out of these 52 SSR markers, 45 were polymorphic, while 7 were monomorphic. The 45 polymorphic markers selected for analysis were distributed across the 12 chromosomes, from those 3 linked to aromatic traits, 4 related to cooking and eating quality traits, and 31 were listed in the panel of 50 standard SSR markers used for marker-trait analysis; the remainder of SSRs were selected randomly.

DNA was extracted from the young leaves of 21-day-old plants using the miniscale method [51]. Each PCR was carried out in a 20 μl reaction volume containing 1 μl of $MgCl_2$ free $10 \times$ PCR buffer with $(NH4)_2SO_4$, 1.2 μl of 25 mM of $MgCl_2$, 0.2 μl of 10 mM of dNTPs, 0.2 μl of 5 U/μl Taq DNA polymerase, 0.5 μl of 10 μM forward and reverse primers, and 3 μl (10 ng) of DNA using a 96-well thermal cycler. An additional 10 μl of mineral oil was added in each well to prevent evaporation. Amplification was carried out using a G-storm PCR machine (Gene Technologies Ltd., England). Amplification conditions were one cycle at 94˚C for 5 minutes (initial denaturation) followed by 35 cycles at 94 ˚C for 1 minute (denaturation), 55 ˚C for 1 minute (annealing), 72 ˚C for 2 minutes (extension), with a final extension for 7 minutes at 72 ˚C at the end of 35 cycles. After mixing with the loading dye, PCR products were run through polyacrylamide gels. A 50 bp DNA ladder was used to determine the amplicon size. Three 4 μl PCR products were resolved by running gel in 1X TB buffer for 1.5 to 2.5 h depending upon the allele size at approximately 90 volts and 500 mA electricity. Gels were then stained with 1 μg/mL of ethidium bromide and documented using a Molecular Imager gel documentation unit (XR System, BIO-RAD, Korea).

### 4.6 Statistical analysis

**Phenotypic analysis.** All data used to analyse and presented in this paper are provided as supplementary files (S1 Fig and S1 Table). The diverse statistical parameters, including mean, standard deviation, coefficient of variation (CV), analysis of variance (ANOVA), frequency distribution and Pearson's correlation coefficient, regression co-efficient of seed Fe, Zn, amylose, and protein contents among accessions were measured using SPSSv17.0. A UPGMA cluster dendrogram of 113 aromatic rice germplasm based on 4 physio-chemical traits was constructed using PAST software package [52] following Ward's hierarchical clustering method.

**Molecular marker-trait association analysis.** The size of the band for each 45 polymorphic markers was scored by AlphaEaseFC 4.0 software. The summary statistics, including the number of alleles, major allele size, and frequency was determined using PowerMarker version 3.25 [53]. The allele frequency data from PowerMarker was used to export the data in binary format (allele presence = "1" and allele absence = "0") for analysis with NTSYS-pc version 2.2 [54]. Association mapping was conducted for four quantitative traits viz., Fe, Zn, amylose, and protein content in rice grain. A general linear model (GLM) was used for assessment of marker-trait associations (MTA) in R version 5.2.2 [55]. The molecular marker-trait associations were calculated using the binary data. To detect the association of Fe, Zn, amylose, and protein content in the rice genome, logarithm of odds (LOD) thresholds were calculated using 1000 permutations with a significance threshold of $p = 0.05$ used as a criteria for QTL analysis of markers with the trait [56]. Here, Composite Interval Mapping (CIM), one of the most frequently used quantitative trait loci (QTL) analysis method [57], was used. Using CIM, putative QTLs were identified using the phenotypic and SSR marker by R/qtl package [58] in R. The QTL map was constructed using MAPdisto version 2.0 [59]. During this analysis, the derived genetic map, missing phenotypes and frequency of other phenotypic data presented as S1 Fig.

## Supporting information

**S1 Data.**
(XLSX)

**S1 Fig. Genetic map, frequency distribution pattern of phenotypic data.**
(DOCX)

**S1 Table. List of 52 SSR markers used in this study.**
(DOC)

## Author Contributions

**Conceptualization:** M. Khalequzzaman.

**Data curation:** M. Z. Islam, S. Banik, M. A. Hossain, J. Ferdous.

**Formal analysis:** M. Z. Islam, M. Arifuzzaman, S. Banik, M. P. Ali.

**Investigation:** M. Z. Islam, M. A. Hossain, M. Khalequzzaman.

**Methodology:** M. Z. Islam.

**Project administration:** M. Khalequzzaman.

**Resources:** M. Khalequzzaman.

**Software:** M. Z. Islam.

**Validation:** B. R. Pittendrigh, M. Tomita, M. P. Ali.

**Visualization:** M. Arifuzzaman, M. P. Ali.

**Writing – original draft:** M. Z. Islam, M. Arifuzzaman, B. R. Pittendrigh, M. Tomita, M. P. Ali.

**Writing – review & editing:** M. Arifuzzaman, B. R. Pittendrigh, M. Tomita, M. P. Ali.

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
