## [Editor Report · Decision Letter 0]

16 Mar 2020

PONE-D-19-31392

Nutritional associated QTLs and diversity analysis in aromatic rice germplasm

PLOS ONE

Dear Dr Ali,

Thank you for submitting your manuscript to PLOS ONE. After careful consideration, we feel that it has merit but does not fully meet PLOS ONE’s publication criteria as it currently stands. Therefore, we invite you to submit a revised version of the manuscript that addresses the points raised during the review process.

We would appreciate receiving your revised manuscript by Apr 30 2020 11:59PM. To enhance the reproducibility of your results, we recommend that if applicable you deposit your laboratory protocols in protocols.io, where a protocol can be assigned its own identifier (DOI) such that it can be cited independently in the future. For instructions see: http://journals.plos.org/plosone/s/submission-guidelines#loc-laboratory-protocols

We look forward to receiving your revised manuscript.

Kind regards,

Evangelia V. Avramidou, PhD

Academic Editor

PLOS ONE

Journal Requirements:

https://pdfs.semanticscholar.org/43d0/358a21b1cd4bb572b3879c77f0cdc9526536.pdf

In your revision ensure you cite all your sources (including your own works), and quote or rephrase any duplicated text outside the methods section. Further consideration is dependent on these concerns being addressed.

4) We note that you have indicated that data from this study are available upon request. PLOS only allows data to be available upon request if there are legal or ethical restrictions on sharing data publicly. For more information on unacceptable data access restrictions, please see http://journals.plos.org/plosone/s/data-availability#loc-unacceptable-data-access-restrictions.

5.Thank you for stating the following financial disclosure:

"The funders had no role in study design, data collection and analysis, decision to

publish, or preparation of the manuscript."

Please provide an amended Funding Statement that declares *all* the funding or sources of support received during this specific study (whether external or internal to your organization) as detailed online in our guide for authors at http://journals.plos.org/plosone/s/submit-now.  

Please state what role the funders took in the study.  If any authors received a salary from any of your funders, please state which authors and which funder. If the funders had no role, please state: "The funders had no role in study design, data collection and analysis, decision to publish, or preparation of the manuscript."

Additional Editor Comments (if provided):

The manuscript entitled: “Nutritional associated QTLs and diversity analysis in aromatic rice germplasm” presents new QTL for aromatic rice varieties in relation with Fe and Zn concentrations. The manuscript, first of all, needs English editing and I also have some major comments below.

Abstract: needs rewriting and English editing

How cluster analysis has been performed? It must be written in the abstract also. How was the QTL were produced?

Line 80: write LSD as Least Square Distance

Line 222: here you report that 45 polymorphic markers were distributed across the 12 chromosomes, how do you know that? Did you perform the analysis or you got this information from previously published papers? I suppose that you want to say that you elected these SSR from the database in order that SSR will be distributed across the 12 chromosomes …etc

Line 257: probability must be indicated with small letter

Line 95: while you discuss about results of cluster analysis, there is no description on materials and methods of the algorithm used for the construction of clusters

Line 137: why this variety had highest concentration, discuss more about this issue.

Line 143: did you have any common genotypes or varieties with other studies?

Line 155: what is GLM analysis?

Line 157: what is GY?

Line 158: what is IRRI?

I cannot understand from the manuscript quite a lot points.

Did you perform association mapping and after CIM for QTL identification? please provide details about the method. This is not clearly described in the materials and methods.

A LOD equal to 2.02 is not a strong LOD in order to state that you found a highly significant QTL,

Also you do not discuss about the variance explained from the QTL which is also low (7.89%), provide examples from other studies with LOD scores and variance explained.

Also you do not discuss about positions of QTL; this is only written in the abstract.
---

## [Author Response · Author response to Decision Letter 0]

5 May 2020

PONE-D-19-31392

Nutritional associated QTLs and diversity analysis in aromatic rice germplasm

Comment: We suggest you thoroughly copyedit your manuscript for language usage, spelling, and grammar. If you do not know anyone who can help you do this, you may wish to consider employing a professional scientific editing service. 

Response: Prof Barry Pittendrigh, Michigan State University, USA edited the paper. 

Additional Editor Comments (if provided):

Comment: The manuscript entitled: “Nutritional associated QTLs and diversity analysis in aromatic rice germplasm” presents new QTL for aromatic rice varieties in relation with Fe and Zn concentrations. The manuscript, first of all, needs English editing and I also have some major comments below.

Response: Thank you so much. The manuscript has been edited by a native English expert Prof Barry Pttendrigh, potential coauthor

Comment: Abstract: needs rewriting and English editing

Response: It has been revised and edited.

Comment: How cluster analysis has been performed? It must be written in the abstract also. How was the QTL were produced?

Response: The cluster analysis and QTL identification procedures are written in the materials and methods section and also briefly added in the abstract. Line 351-361, Line 340-342. 

Comment: Line 80: write LSD as Least Square Distance

Response: Thanks, we elaborated the full meaning of LSD that indicate in our paper. LSD indicates Least Significant Difference. This full name has been added in the text.

Comment: Line 222: here you report that 45 polymorphic markers were distributed across the 12 chromosomes, how do you know that? Did you perform the analysis or you got this information from previously published papers? I suppose that you want to say that you elected these SSR from the database in order that SSR will be distributed across the 12 chromosomes …etc

Response: Yes, you are right. We get these SSR information from Gramene website (https://archive.gramene.org/markers/microsat/). This can be found in the reference number 44.

Comment: Line 257: probability must be indicated with small letter

Response: Okay, thank you. The P is corrected with p in the text.

Comment: Line 95: while you discuss about results of cluster analysis, there is no description on materials and methods of the algorithm used for the construction of clusters

Response: Thank you very much for your nice comment. Now we added the method which was used to construct cluster. A UPGMA cluster dendrogram of 113 aromatic rice germplasm based on 4 physio-chemical traits was constructed using PAST software package (46) following Ward's hierarchical clustering method. Line 340-342.

Comment: Line 137: why this variety had highest concentration, discuss more about this issue.

Response: Thank you very much this comment. We discussed this issue elaborately in discussion section. Line 170-177.

Comment: Line 143: did you have any common genotypes or varieties with other studies?

Response: Yes. We used common germplasm which was already used in other studies both phenotypic and genotypic analysis (Islam et al. 2017). In this study, we used four quantitative physico-chemical data (Fe, Zn, protein and amylose content) only which were not published anywhere. 

Islam MZ, M Khalequzzaman, M K Bashar, N A Ivy, MA K Mian, BR Pittendrigh, MM Haque and MP Ali (2018) Variability assessment of aromatic rice germplasm by pheno-genomic traits and population structure analysis. Scientific Reports volume 8, Article number: 9911.

Comment: Line 155: what is GLM analysis?

Response: GLM stands for General linear model. This model was used for assessment of marker-trait associations (MTA). Line 353.

Comment: Line 157: what is GY?

Response: GY stands grain yield. Corrected in the text.

Comment: Line 158: what is IRRI?

Response: IRRI stands for International Rice Research Institute. Now added in text.

Comment: I cannot understand from the manuscript quite a lot points. Did you perform association mapping and after CIM for QTL identification? please provide details about the method. This is not clearly described in the materials and methods.

Response: We performed association mapping for four quantitative traits viz., Fe, Zn, amylose and protein content in rice grain. We also performed CIM for QTLs identification and map was constructed using MAPdisto version 2.0 [51]. Line 352-361.

Comment: A LOD equal to 2.02 is not a strong LOD in order to state that you found a highly significant QTL, Also you do not discuss about the variance explained from the QTL which is also low (7.89%), provide examples from other studies with LOD scores and variance explained.

Response: Thank for this important comment which strongly improve our manuscript. We added this information in discussion section. Lines 222-244.

Comment: Also you do not discuss about positions of QTL; this is only written in the abstract.

Response: Thanks for this comment. Now we have added the position of all QTLs in the discussion part. Line 143-150. Line 218-234.

---

## [Editor Report · Decision Letter 1]

18 May 2020

PONE-D-19-31392R1

Mapping QTLs underpin nutrition components in aromatic rice germplasm

PLOS ONE

Dear Dr Ali,

Thank you for submitting your manuscript to PLOS ONE. After careful consideration, we feel that it has merit but does not fully meet PLOS ONE’s publication criteria as it currently stands. Therefore, we invite you to submit a revised version of the manuscript that addresses the points raised during the review process.

We would appreciate receiving your revised manuscript by Jul 02 2020 11:59PM. To enhance the reproducibility of your results, we recommend that if applicable you deposit your laboratory protocols in protocols.io, where a protocol can be assigned its own identifier (DOI) such that it can be cited independently in the future. For instructions see: http://journals.plos.org/plosone/s/submission-guidelines#loc-laboratory-protocols

We look forward to receiving your revised manuscript.

Kind regards,

Evangelia V. Avramidou, PhD

Academic Editor

PLOS ONE

Additional Editor Comments (if provided):

Dear authors,

I have still quite points in order to improve your manuscript. First of all the revised manuscript did not have numbered lines so I had to work in the pdf file by adding comments.

Please adress all the comments that I have included in the attached pdf.

---

## [Author Response · Author response to Decision Letter 1]

20 May 2020

Response of Reviewer comments

Mentioned line number of revised texts matched with only track changed revised manuscript. So please match with track changed version for line numbering.

1. Rephrase as: “As rice is an important staple food globally, research for development and enhancement of its nutritional value it is an imperative task.”

Response: We revised as suggested sentence. Line 26-27.

2. You cannot say healthier rice maybe you should write “better quality”.

Response: We rephrased as better quality at each location throughout the manuscript. Line 29.

3. In this study, we analyzed 113 aromatic rice germplasm in order to identify quantitative trait loci (QTL) underpinning nutrition components and measured by measuring the normal frequency distribution for Fe, Zn, amylose, and protein content in those rice germplasm.

Response: We revised as suggested sentence. Line 29-32

4. Erase “in the tested germplasm.”

Response: We erased these. Line 35.

5. Please rephrase the sentence

Response: We revised it. Line 50-51.

6. position instead of positions

Response: We deleted “s” from positions. Line 51

7. substitute release with development of new varieties

Response: We did it. Line 57-58.

8. please provide also LOD scores in the abstract

Response: We added LOD value where necessary. Thanks

9. provide a reference (Introduction start)

Response: We added reference 6 (Richard 2012).

10. Erase “Given that” substitute with “According to FAO, more than…”

Response: We did it. Line 81

11. erase thus substitute with for

Response: We did. Line 85.

12. Initiate another sentence from: “as has already been done for already Zn-enriched rice varieties in Bangladesh, India, and Philippines [12].” But rewrite it as: “A biofortification breeding program was succeeded for Zn-enriched rice varieties in Bangladesh, India, and Philippines [12].”

Response: We revised as suggested sentence. Line 88-89.

13. rewrite the sentence

Response: We revised this sentence. Line 96-98.

14. better quality rice in terms of a healthier life

Response: We changed this. Line 122

15. Erase: “including but not limited to” and substitute with “like complex genetic….”

Response: We used like. Line 125

16. erase "too"

Response: We erased “too”.

17. Rephrase “In contrast, to identify” as “In contrast, identification of quantitative trait loci (QTLs)…”

Response: We did id. Line 128

18. Substitute: “Further” with “Furthermore,”

Response: We changed it. Line 130.

19. Rephrase “few to no studies” with “limited studies”

Response: We did it. Line 149

20. Substitute “high yielding genotypes for Fe” with “high yield genotypes for Fe”

Response: We did it. Line 151

21. specify the units (results start)

22. specify the units

Response: We added units where necessary. Thanks.

23. Exhibited

Response: We changed as exhibited. Line 167

24. specify the units

25. provide LOD score

26. provide LOD scores

27. provide LOD score

28. provide LOD scores

Response: We added LOD value and units where necessary. Thanks

29. provide a reference (discussion start)

Response: We added references (30, 31). Line 232.

30. why did you use Mahalanobis distance explain the superiority over other genetic distances

Response: Data fitted well. So we used this distance.

31. I do not agree with the term major due to two reasons A major QTL is when LOD score is above 3 and major effects they have if they explain a high percentage of variance, here you report one QTL with a maximum LOD 2.02 and 7.89 % variance

Response: We revised this such as we identified several novel loci. Line 256

32. replace minor instead of major

Response: Thanks for this important comment. We changed it. Line 260

33. LOD value?

Response: We added LOD value. Line 264

34. place a "="

Response: We added =. Line 269.

35. replace researchers with studies

Response: We replaced with studies. Line 271.

36. Add " In the same study, another locus..."

Response: We added suggested phrase. Line 280

37. add "QTLS"

Response: We added. Line 288

38. erase we used (Mathods)

Response: We erased them. Line 291

39. add were used

Response: We added them. Line 293

40. 52 instead of fifty-two

Response: We wrote 52 instead of fifty-two. Line 343

41. erase in terms of their bands

Response: We erased these. Line 345

42. I think that it is important to show to to the readers which SSR markers you used so please provide a supplementary table

Response: We added markers data as supplementary file. Line 343.

43. Erase “with” 3 were linked to aromatic traits,” and replace it with “from those 3 were linked

Response: We did it. Line 347.

44. erase were 

Response: Thanks. We erased were. Line 370.

---

## [Editor Report · Decision Letter 2]

21 May 2020

PONE-D-19-31392R2

Mapping QTLs underpin nutrition components in aromatic rice germplasm

PLOS ONE

Dear Dr. Ali,

Thank you for submitting your manuscript to PLOS ONE. After careful consideration, we feel that it has merit but does not fully meet PLOS ONE’s publication criteria as it currently stands. Therefore, we invite you to submit a revised version of the manuscript that addresses the points raised during the review process.

We look forward to receiving your revised manuscript.

Kind regards,

Evangelia V. Avramidou, PhD

Academic Editor

PLOS ONE

Additional Editor Comments (if provided):

Dear authors,

I have a single correction in Line 260 substitute "we used" with "were used". Please correct it.

---

## [Author Response · Author response to Decision Letter 2]

21 May 2020

Commend: Dear authors,

I have a single correction in Line 260 substitute "we used" with "were used". Please correct it. 

Response: We revised it. Thanks. Line 254.

---

## [Editor Report · Decision Letter 3]

27 May 2020

Mapping QTLs underpin nutrition components in aromatic rice germplasm

PONE-D-19-31392R3

Dear Dr. Ali,

We are pleased to inform you that your manuscript has been judged scientifically suitable for publication and will be formally accepted for publication once it complies with all outstanding technical requirements.

With kind regards,

Evangelia V. Avramidou, PhD

Academic Editor

PLOS ONE

Additional Editor Comments (optional):

Dear authors,

I think that now, manuscript is ready for publication.

With kind regards
---

## [Editor Report · Acceptance letter]

1 Jun 2020

PONE-D-19-31392R3 

Mapping QTLs underpin nutrition components in aromatic rice germplasm 

Dear Dr. Ali:

I am pleased to inform you that your manuscript has been deemed suitable for publication in PLOS ONE. Congratulations! Your manuscript is now with our production department. 

With kind regards,

on behalf of

Dr. Evangelia V. Avramidou 

Academic Editor

PLOS ONE